# Effectiveness of kaolin-impregnated hemostatic gauze use in preperitoneal pelvic packing for patients with pelvic fractures and hemodynamic instability: A propensity score matching analysis

**Kwangmin Kim** [1,2], **Hongjin Shim** [1,2], **Pil Young Jung** [1,2], **Seongyup Kim** [1,2], **Young Un Choi** [1,2], **Keum Seok Bae** [1,2], **Jung Kuk Lee** [3], **Ji Young Jang** [4] *

1 Department of Surgery, Yonsei University Wonju College of Medicine, Wonju, Korea, 2 Regional Trauma Center, Wonju Severance Christian Hospital, Wonju, Korea, 3 Department of Biostatistics, Yonsei University Wonju College of Medicine, Wonju, Korea, 4 Department of Surgery, Trauma Center, National Health Insurance Service Ilsan Hospital, Goyang, Korea

* drjangjiyoung@gmail.com, jyjang@hanmail.net

**Data Availability Statement:** All relevant data are within the paper and its Supporting Information files.

## Abstract

### Introduction

We evaluated the effectiveness of kaolin-impregnated hemostatic gauze use in preperitoneal pelvic packing (PPP) for patients with hemodynamic instability due to severe pelvic fractures.

### Materials and methods

Between May 2014 and October 2018, 53 of 75 patients who underwent PPP due to hemodynamic instability induced by pelvic fracture were enrolled. Their medical records were prospectively collected and retrospectively analyzed. QuikClot combat gauze (hydrophilic gauze impregnated with kaolin) and general surgical tape were used in 21 patients, while general surgical tape was used in the remaining 32 patients.

### Results

As there were differences in the characteristics of patients between the hemostatic gauze (HG) group and control group, propensity score matching (PSM) was performed to adjust for age, sex, and lactate levels. After PSM, the clinical characteristics between the two groups became similar. There were no differences in the rates of mortality and hemorrhage-induced mortality between the two groups. However, the packed red blood cell (RBC) requirement for an additional 12 hours in the HG group was significantly lower than that in the control group (4.1 ± 3.5 vs. 7.6 ± 6.1 units, p = 0.035). The lengths of intensive care unit and hospital stays tended to be shorter in the HG group than in the control group (11.6 vs. 18.5 days, p = 0.1582; 30.8 vs. 47.4 days, p = 0.1861, respectively).

**Funding:** The authors received no specific funding for this work.

**Competing interests:** The authors have declared that no competing interests exist.

**Abbreviations:** AP, anteroposterior; APC, anteroposterior compression; EF, external fixator; ER, emergency room; HG, hemostatic gauze; ICU, intensive care unit; PPP, preperitoneal pelvic packing; PSM, propensity score matching; QCG, QuikClot Combat Gauze; RBCs, red blood cells; SI, sacroiliac.

## Conclusions

The use of HG during PPP did not reduce hemorrhage-induced mortality, but did reduce the need for additional packed RBC transfusions in patients with hemodynamic instability due to severe pelvic fractures.

## 1 Introduction

The mortality rate in patients with hemodynamic instability due to pelvic bone fractures remains high despite the development of several hemostatic modalities [1–5]. Recent studies have shown that preperitoneal pelvic packing (PPP) can be an effective hemostatic method for damage control surgery [6–8].

Among commercially available hemostatic gauzes, QuikClot Combat Gauze (QCG; Z-Medica, Wallingford, CT, USA), a hydrophilic gauze impregnated with kaolin, enhances hemostasis by activating the intrinsic pathway. It was first used in trauma cases with hemorrhaging external wounds but has recently been used for patients with intracorporeal hemorrhage [9, 10].

A study using a hypothermic coagulopathic swine model showed that, compared to plain gauze, kaolin-impregnated gauze for packing in cases of high-grade liver injuries reduced postoperative hemorrhage [11].

The present study aimed to evaluate the effectiveness of kaolin-impregnated hemostatic gauze in PPP for patients with hemodynamic instability due to severe pelvic fractures.

## 2 Material and methods

### 2.1 Patient selection and data collection

This retrospective study was approved by the institutional review board of a tertiary university hospital (IRB no. CR319078). Medical data of patients with pelvic fractures were collected from the hemodynamically unstable pelvic bone fracture registry of this tertiary university hospital, which is part of the Korean Trauma Data Bank. Data were collected prospectively and analyzed retrospectively. Because the data were analyzed anonymously, informed consent was waived. The inclusion criteria were: 1) hemodynamically unstable pelvic fracture, and 2) age > 19 years. Seventy-five patients with hemodynamic instability due to pelvic fractures who were admitted to the regional trauma center of the tertiary university hospital between May 2014 and October 2018 were enrolled in the study. After the exclusion of 22 patients who did not receive PPP, 53 patients were included. These patients were divided into a hemostatic gauze group (HG group, n = 22) and a control group (n = 31) (Fig 1). Hemostatic gauze was commonly used as a standard of care in our hospital. When performing PPP, there were no criteria for the use of hemostatic gauze and its use was determined by the surgeon.

### 2.2 Patient management

Hemodynamic instability was defined as persistent hypotension (systolic blood pressure < 90 mmHg), even after 2 L of crystalloid loading or transfusion of 2 units of packed red blood cells (RBCs). When patients were admitted to the trauma bay with pelvic bone fractures and hemodynamic instability, they underwent extended focused assessment with sonography for trauma and a trauma series of X-rays (lateral cervical spine, chest anteroposterior [AP], and pelvis AP). Thoracoabdominal injuries were evaluated using the images. Based on the results,

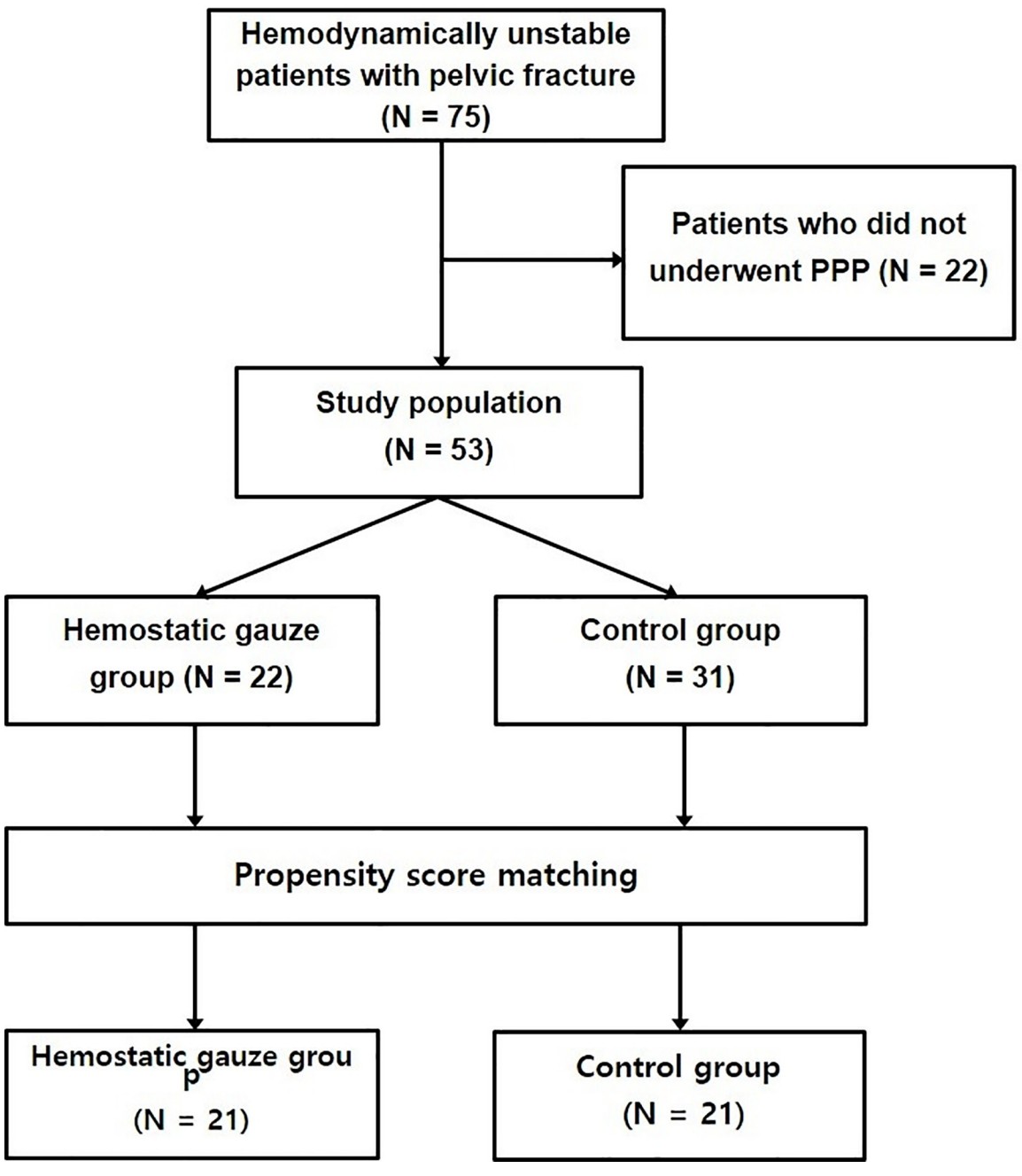

**Fig 1. Study flow chart.** PPP, preperitoneal pelvic packing.

emergency thoracotomy or laparotomy was performed. After pelvic binders were applied to reduce pelvic volume, PPP was performed for patients with pelvic ring injuries based on pelvic AP imaging results. Orthopedic surgeons on the trauma team then determined whether external fixation of the pelvic fractures should be performed. The management protocol for hemodynamically unstable pelvic fractures was initiated in our trauma center in May 2014 according to management protocols created by the Rocky Mountain Regional Trauma Center at Denver Health [12].

Following the protocol, secondary pelvic angiography following PPP was performed in cases of ongoing bleeding even if PPP was performed. Embolization procedures were performed in cases of contrast media extravasations during pelvic angiography. In all pelvic fracture patients with shock, pelvic binders were applied in the Emergency Room (ER) and then were removed after patients became hemodynamically stable. In cases requiring external fixator (EF) application, pelvic binders were not re-applied; rather, these were re-applied just after PPP, but only in cases without EF use. In our institution, C-clamp was not used and supra-acetabular EF was performed for all pelvic external fixations. Prophylactic antibiotics were administered preoperatively to patients who underwent PPP and were continued until the second look.

## 2.3 PPP techniques and use of hemostatic gauze

PPP was performed by trauma surgeons who successfully completed Definitive Surgical Trauma Care (DSTC[TM]) training provided by the International Association for Trauma Surgery and Intensive Care. After creating a 7–8 cm vertical skin incision beginning at the pubis symphysis, the anterior sheath of the rectus abdominis muscle was resected and the muscle was split. After the preperitoneal space was bluntly dissected in the posterolateral direction and the peritoneum migrated to the medial side, the lower border of the sacroiliac (SI) joint was examined. Three surgical pads were sequentially packed from the lower border of the SI joint using ringed forceps. This procedure was repeated on the contralateral side. After coagulopathy, hypothermia, and metabolic acidosis were corrected, packed surgical pads were removed and the abdominal wall was repaired within 48 hours [13]. One kaolin-impregnated gauze (QCG) and two surgical pads were used for patients in the HG group. After one QCG was packed into the lower border of the SI joint, two surgical pads were also packed. This procedure was repeated on the contralateral side. Patients in the control group were treated with the same procedure; however, the packing consisted of three plain surgical pads and no QCG.

## 2.4 Kaolin-impregnated hemostatic gauze

QCG is currently the Committee on Tactical Combat Casualty Care-recommended standard hemostatic agent of the United States military. It is a nonwoven surgical gauze coated in kaolin (aluminosilicate clay) that activates the intrinsic coagulation pathway. QCG has equal or higher efficacy in laboratory tests than other hemostatic agents including TraumaStat (Ore-Medix, Salem, OR, USA), Celox-D (SAM Medical, Portland, OR, USA), and HemCon RTS bandage (HemCon, Portland, OR, USA). QCG appeared to produce no short-term vascular damage compared with standard gauze in an animal model, and no adverse reactions during its use on the battlefield during Operation Cast Lead in the Gaza Strip [14].

## 2.5 Outcome evaluations

The primary study outcome was the rate of hemorrhage-induced mortality. The secondary outcomes were requirement of packed RBCs (during the initial 4 hours and after an additional 12 hours), length of intensive care unit (ICU) stay, and length of hospital stay.

## 2.6 Statistical analysis

Continuous variables are presented as mean ± standard deviation or median (range). The chi-square test, Student's t-test, Fisher's exact test, and Mann-Whitney U test were used to compare the groups. For propensity score matching (PSM), logistic regression analysis was performed for all baseline features that differed between the HG group and control group on

multivariate analysis. A propensity score for the predicted probability of a patient using hemostatic gauze was estimated using a multivariable logistic regression model. The C-statistic of the logistic regression model for PSM was 0.703. Adjusted covariates in PSM included age (>75 years), sex, and highest lactate level (>4 mmol/L). Using the nearest neighbor matching method, the absolute values of the differences in the estimated propensity scores of all patients in the HG group and control group were paired from smallest to largest. P values < 0.05 were considered statistically significant. All calculations were performed using SAS 9.4 (SAS Institute, Cary, NC, USA).

## 3 Results

### 3.1 Clinical characteristics of patients with hemodynamic instability due to pelvic fractures

The mean patient age was 60.7 ± 17.9 years; 32 patients (60.4%) were male. The mean injury severity score was 40.2 ± 11.0 and the highest serum lactate level at the time of arrival at the ER was 6.74 ± 4.13 mmol/L. According to the Young-Burgess classification of pelvic fracture types, 13 patients (24.5%) had lateral compression fracture type III, 17 (32.1%) had lateral compression fracture type II, and 16 (30.2%) had vertical shearing. Four patients (7.5%) had anteroposterior compression (APC) type III, while 2 patients (3.8%) had APC type II. Pelvic angioembolizations were performed in 12 patients (22.6%). Hemostatic gauze was used for 21 patients (39.6%). A total of 21.8 ± 11.6 units of packed RBCs were required. The mean length of ICU was 12 days. There were 24 cases of mortality (45.3%); among them, 11 were hemorrhagic (20.8%) (Table 1).

### 3.2 HG group vs. control group before PSM

The comparison of the 21 patients in the HG group and the 32 patients in the control group revealed no significant difference in age. However, the ratio of patients >75 years of age was significantly different (38.1% vs. 12.5%, p = 0.045). The mean value of the highest lactate level in each group was not different (5.7 ± 3.7 vs. 7.4 ± 4.3 mmol/L, p = 0.150). The ratio of patients who had lactate levels > 4 mmol/L in the control group was significantly higher than of patients who had lactate levels > 4 mmol/L in the HG group (84.4% vs. 57.1%, p = 0.028). The other variables in the groups were not different (Table 2). Comparison of the clinical outcomes revealed no significant intergroup differences (Table 3).

### 3.3 HG group vs. control group after PSM

One-to-one PSM was performed for three variables that showed statistical differences (age > 75 years, serum lactate > 4 mmol/L, and sex). Comparison of the 21 patients from both groups using PSM identified no additional significant differences. In other words, the basic clinical variables for both groups were properly revised (Table 4). When clinical outcome was compared between the HG and the control groups, mortality due to hemorrhage (23.8% vs. 19.1%, p = 1.00) did not differ significantly. However, significantly lower amounts of packed RBCs were required in the HG group for an additional 12 hours than for the control group (4.1 ± 3.5 vs. 7.6 ± 6.1 units, p = 0.035). Although there were no statistically significant intergroup differences in length of hospital and ICU stays, both variables tended to be shorter in the HG group (length of ICU stay, 11.6 vs. 18.5 days, p = 0.158; length of hospital stay, 30.8 vs. 47.4 days, p = 0.186) (Table 5).

**Table 1. Patient characteristics.**

|  | n = 53 |
|---|---|
| Age (years) | 60.7 ± 17.9 |
| Sex (male) | 32 (60.4%) |
| ISS | 40.2 ± 11.0 |
| Cardiac arrest in ER | 7 (13.2%) |
| DM | 9/52 (17.3%) |
| Anticoagulant use | 7/53 (13.5%) |
| Initial SBP (mmHg) | 82.0 ± 34.7 |
| Initial hemoglobin (g/dL) | 10.0 ± 2.8 |
| Worst lactate level in ER (mmol/L) | 6.74 ± 4.13 |
| Combined injury | 50 (94.3%) |
| Pelvic fracture type | |
| APC type II | 2 (3.8%) |
| APC type III | 4 (7.5%) |
| LC I | 1 (1.9%) |
| LC II | 17 (32.1%) |
| LC III | 13 (24.5%) |
| VS | 16 (30.2%) |
| Time to PPP from ER arrival (mins) | 87 (26–555) |
| Pelvic external fixation | 10 (18.9%) |
| Emergent pelvic angiography | 12 (22.6%) |
| Concurrent laparotomy | 13 (24.5%) |
| Hemostatic gauze use | 21 (39.6%) |
| Packed RBC requirement for 4h (units) | 12.0 ± 9.6 |
| Packed RBC requirement for 12h (units) | 5.4 ± 5.1 |
| Total packed RBC requirement (units) | 21.8 ± 11.6 |
| Time to tape removal (hour) | 45.5 ± 21.1 |
| Duration of hospitalization (day) | 29 (1–257) |
| Duration of ICU stay (day) | 12 (1–255) |
| Mortality | 24 (45.3%) |
| Mortality due to hemorrhage | 11 (20.8%) |

ISS: Injury severity score, ER: emergency room, DM: Diabetes mellitus, SBP: Systolic blood pressure, APC: Anterior-posterior compression, LC: Lateral compression, VS: Vertical shear, PPP: Pre-peritoneal pelvic packing, RBC: Red blood cells, ICU: Intensive care unit.

## 4 Discussion

In this study, the mean injury severity score was 40.2 ± 11.0; 13.2% of patients were in cardiac arrest on arrival at the ER, and the mean lactate level in the ER was 6.74 mmol/L. Therefore, patients enrolled in this study were hemodynamically unstable and may have had severe coagulopathy. Since coagulopathy can be aggravated by hypothermia and metabolic acidosis in patients with hemorrhagic shock, we hypothesized that the difference in the ratio of patients with a lactate level > 4 mmol/L between the HG group and control group in this study influenced the amount of hemorrhage in both groups. A study on patients with pelvic bone fractures showed that hypothermia and elevated serum lactate levels were predictors of hemorrhage control interventions [15]. In addition, in the present study, the number of patients >75 years old differed significantly between the two groups. Therefore, these three variables (age, sex, and lactate level) were independent variables in PSM to compensate for

**Table 2. Hemostatic gauze group vs. control group before propensity score matching.**

|  | Hemostatic gauze group n = 21 | Control group n = 32 | p-value |
|---|---|---|---|
| Age (years) | 65.4 ± 19.5 | 57.7 ± 16.4 | 0.126 |
| Age > 75 | 8 (38.1%) | 4 (12.5%) | 0.045* |
| Sex (male) | 13 (61.9%) | 19 (59.4%) | 0.854 |
| ISS | 38.2 ± 9.8 | 41.5 ± 11.7 | 0.296 |
| Cardiac arrest in ER | 2 (9.5%) | 5 (15.6%) | 0.690* |
| DM | 2 (9.5%) | 7 (22.6%) | 0.283* |
| Anticoagulant use | 4 (19.0%) | 3 (9.7%) | 0.420* |
| Initial SBP | 78.9 ± 36.5 | 84.1 ± 33.8 | 0.598 |
| Initial SBP < 90 mmHg | 15 (71.4%) | 20 (62.5%) | 0.502 |
| Initial hemoglobin (g/dL) | 10.3 ± 2.7 | 9.8 ± 2.9 | 0.478 |
| Worst lactate level in ER (mmol/L) | 5.7 ± 3.7 | 7.4 ± 4.3 | 0.150 |
| Worst lactate level in ER > 4 (mmol/L) | 12 (57.1%) | 27 (84.4%) | 0.028 |
| Combined injury | 18 (85.7%) | 32 (100%) | 0.057 |
| Pelvic fracture type |  |  | 0.986* |
| APC type II | 1 (4.8%) | 1 (3.1%) |  |
| APC type III | 1 (4.8%) | 3 (9.4%) |  |
| LC I | 0 | 1 (3.1%) |  |
| LC II | 7 (33.3%) | 10 (31.3%) |  |
| LC III | 5 (23.8%) | 8 (25.0%) |  |
| VS | 7 (33.3%) | 9 (28.1%) |  |
| Pelvic external fixation | 4 (19.0%) | 6 (18.8%) | 1.000* |
| Emergent pelvic angiography | 5 (23.8%) | 7 (21.9%) | 0.869 |

*Result of Fisher's exact test

ISS: Injury severity score, ER: emergency room, DM: Diabetes mellitus, SBP: Systolic blood pressure, APC: Anterior-posterior compression, LC: Lateral compression, VS: Vertical shear.

differences in patient characteristics between the HG group and control group. In the present study, although there was no difference in hemorrhagic mortality rates between the compensated HG group and the control group, the number of transfusions for an additional 12 hours in the HG group was significantly lower than that in the control group.

After the use of kaolin-impregnated gauze was initially reported in the military setting to stop bleeding due to external wounds, hemostatic effects started being reported in civilian

**Table 3. Comparison of clinical outcomes between the hemostatic gauze group and the control group before propensity scoring matching.**

|  | Hemostatic gauze group n = 21 | Control group n = 32 | p-value |
|---|---|---|---|
| Packed RBC requirement for 4h (units) | 11.6 ± 6.5 | 12.3 ± 11.2 | 0.785 |
| Packed RBC requirement for 12h (units) | 4.1 ± 3.5 | 6.3 ± 5.9 | 0.094 |
| Total packed RBC requirement (units) | 20.0 ± 9.3 | 23.1 ± 12.9 | 0.346 |
| Time to tape removal (hours) | 52.6 ± 20.1 | 41.5 ± 20.9 | 0.116 |
| Duration of hospitalization (days) | 26 (1–129) | 30 (1–257) | 0.084 |
| Duration of ICU stay (days) | 7 (1–40) | 12.5 (1–255) | 0.204 |
| Mortality | 12 (57.1%) | 12 (37.5%) | 0.160 |
| Mortality due to hemorrhage | 5 (23.8%) | 6 (18.8%) | 0.657 |

RBC: Red blood cells, ICU: Intensive care unit

*Result of Fisher's exact test

**Table 4. Hemostatic gauze group vs. control group after PSM.**

|  | Hemostatic gauze group n = 21 | Control group n = 21 | p-value |
|---|---|---|---|
| Age (years) | 65.4 ± 19.5 | 59.1 ± 18.1 | 0.289 |
| Age > 75 | 8 (38.1%) | 4 (19.1%) | 0.306 |
| Sex (male) | 13 (61.9%) | 14 (66.7%) | 1.000 |
| ISS | 38.2 ± 9.8 | 40.1 ± 9.8 | 0.532 |
| Cardiac arrest in ER | 2 (9.5%) | 2 (9.5%) | 1.000* |
| DM | 2 (9.5%) | 6 (30.0%) | 0.130* |
| Anticoagulant use‡ | 4 (19.0%) | 3 (15.0%) | 1.000* |
| Initial SBP | 78.9 ± 36.5 | 97.6 ± 30.3 | 0.077 |
| Initial SBP < 90 mmHg | 15 (71.4%) | 10 (47.6%) | 0.208 |
| Initial hemoglobin (g/dL) | 10.3 ± 2.7 | 10.0 ± 2.7 | 0.750 |
| Worst lactate level in ER (mmol/L) | 5.7 ± 3.7 | 6.4 ± 3.5 | 0.525 |
| Worst lactate level in ER > 4 (mmol/L) | 12 (57.1%) | 16 (76.2%) | 0.326 |
| Combined injury | 18 (85.7%) | 21 (100%) | 0.232 |
| Pelvic fracture type |  |  | 0.676* |
| APC type II | 1 (4.8%) | 1 (4.8%) |  |
| APC type III | 1 (4.8%) | 2 (9.5%) |  |
| LC I | 0 | 1 (4.8%) |  |
| LC II | 7 (33.3%) | 8 (38.1%) |  |
| LC III | 5 (23.8%) | 5 (23.8%) |  |
| VS | 7 (33.3%) | 4 (19.1%) |  |
| Pelvic external fixation | 4(19.0%) | 4(19.0%) | 1.000* |
| Emergent pelvic angiography | 5 (23.8%) | 4 (19.0%) | 1.000* |

‡data missing = 1, Results of Fisher's exact test

ISS: Injury severity score, ER: emergency room, DM: Diabetes mellitus, SBP: Systolic blood pressure, APC: Anterior-posterior compression, LC: Lateral compression, VS: Vertical shear, PSM: propensity score matching.

cohorts [9, 16], while some animal studies showed effective intracorporeal hemostasis using this type of gauze [11, 14]. Inaba et al. reported that the amount of bleeding during gauze packing was significantly lower in the kaolin-impregnated hemostatic gauze group than in the standard gauze group in a randomized controlled animal trial using a damage-control swine model with grade IV liver injury [17]. However, the number of human studies was limited, with very few on intracorporeal use with this gauze type [10, 18, 19].

**Table 5. Comparison of clinical outcomes between hemostatic gauze group and control group after PSM.**

|  | Hemostatic gauze group n = 21 | Control group n = 21 | p-value |
|---|---|---|---|
| Packed RBC requirement for 4h (units) | 11.6 ± 6.5 | 12.1 ± 12.3 | 0.852 |
| Packed RBC requirement for 12h (units) | 4.1 ± 3.5 | 7.6 ± 6.1 | 0.035 |
| Total packed RBC requirement (units) | 20.0 ± 9.3 | 24.3 ± 14.3 | 0.252 |
| Time to tape removal (hours) | 52.6 ± 20.1 | 40.4 ± 22.7 | 0.127 |
| Duration of hospitalization (days) | 30.8 (1–129) | 47.4 (1–130) | 0.186 |
| Duration of ICU stay (days) | 11.6 (1–40) | 18.5 (1–76) | 0.158 |
| Mortality | 12 (57.1%) | 8 (38.1%) | 0.354 |
| Mortality due to hemorrhage | 5 (23.8%) | 4 (19.1%) | 1.000 |

RBC: Red blood cells, ICU: Intensive care unit, PSM: propensity score matching.

Choron et al. showed that the use of kaolin-impregnated hemostatic gauze and laparotomy pads did not differ from standard packing in terms of the amount of intraoperative transfusions and complications that developed by intra-abdominal packing, although the group using kaolin-impregnated hemostatic gauze and laparotomy pads in damage-control laparotomy had more severe physiologic derangement than did the control group [19]. Additionally, the incidence of infectious complications such as wound infection, dehiscence, and intra-abdominal abscess was not statistically different between the two groups [13].

Uncontrolled hemorrhage remains a major cause of mortality and morbidity in military and civilian trauma [20]. In our study, hemorrhagic mortality rates did not differ significantly between the two groups; however, the HG group (5, 23.8%) showed a slightly higher rate than the control group (4, 19.1%) (p = 1.000). The use of QCG alone would limit the reduction of hemorrhagic mortality for the enrolled patients because of the severity of the hemorrhages.

Kaolin-impregnated hemostatic gauze promotes clotting by activating factors XII and XI in the intrinsic coagulation pathway [20]. A recent prospective experimental study used kaolin-impregnated hemostatic gauze in a hypothermic swine model. The test group, which was intentionally hemodiluted with large amounts of IV fluid, showed significantly less bleeding than the control group. These results suggest that kaolin-impregnated hemostatic gauze can be used effectively in coagulopathy-induced traumatic hemorrhagic patients [21]. This outcome was similar to that in the present study in which the transfusion volume for an additional 12 hours was lower in the HG group than in the control group. In our research, transfusions were performed within 4 hours before and during surgery, with no significant intergroup differences during the first 4 hours. These data, however, did not assist with the evaluation of the hemostatic effect of kaolin-impregnated hemostatic gauze. Conversely, since blood transfusion volumes for an additional 12 hours were determined by coagulopathy and bony and venous bleeding after damage control surgery, the volumes during this period might indirectly reflect the hemostatic effect of packed kaolin-impregnated hemostatic gauze.

This study has some limitations. It was a single-center retrospective study and included a small number of patients. Therefore, PSM was possible using only 3 variables. Despite these limitations, this study is meaningful since it showed that the use of kaolin-impregnated hemostatic gauze as a packing material during PPP in patients with hemodynamic instability due to pelvic fractures reduced the amount of blood loss for an additional 12 hours without an increase in postoperative wound infections. Further randomized control studies on this topic are needed.

## 5 Conclusions

Hemostatic gauze during PPP can be effectively used to reduce blood loss for patients with hemodynamic instability due to severe pelvic fractures.

## Acknowledgments

We thank the staff members of the regional trauma centers in Korea for their enthusiasm and commitment to patient care.

## Author Contributions

**Conceptualization:** Hongjin Shim, Ji Young Jang.

**Data curation:** Kwangmin Kim, Pil Young Jung, Seongyup Kim, Young Un Choi, Ji Young Jang.

**Formal analysis:** Jung Kuk Lee, Ji Young Jang.

**Investigation:** Kwangmin Kim, Ji Young Jang.

**Methodology:** Kwangmin Kim, Ji Young Jang.

**Supervision:** Keum Seok Bae, Ji Young Jang.

**Visualization:** Jung Kuk Lee.

**Writing – original draft:** Kwangmin Kim, Ji Young Jang.

**Writing – review & editing:** Hongjin Shim, Pil Young Jung, Seongyup Kim, Young Un Choi, Keum Seok Bae, Ji Young Jang.

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
