## [Decision Letter · Decision Letter 0]

3 Apr 2020

PONE-D-19-34531

Effectiveness of kaolin-impregnated hemostatic gauze use in preperitoneal pelvic packing for patients with pelvic fractures and hemodynamic instability: propensity score matching analysis

PLOS ONE

Dear Dr. Jang,

Thank you for submitting your manuscript to PLOS ONE. After careful consideration, we feel that it has merit but does not fully meet PLOS ONE’s publication criteria as it currently stands. Therefore, we invite you to submit a revised version of the manuscript that addresses the points raised during the review process.

The authors are required to respond to the reviewer's comments and to add a brief about their standard of care of patients with pelvic ring injury associated with haemodnamic instability. This will explain the use of HG in the study group and the role of angio-embolization.Although the authors discussed the small sample size as a limitation of the study, they did not mention the cause of including only 53 out of 75 patients with PPP.

We would appreciate receiving your revised manuscript by May 18 2020 11:59PM. To enhance the reproducibility of your results, we recommend that if applicable you deposit your laboratory protocols in protocols.io, where a protocol can be assigned its own identifier (DOI) such that it can be cited independently in the future. For instructions see: http://journals.plos.org/plosone/s/submission-guidelines#loc-laboratory-protocols

We look forward to receiving your revised manuscript.

Kind regards,

Osama Farouk

Academic Editor

PLOS ONE

Journal Requirements:

2. Please provide additional details regarding participant consent. In the ethics statement in the Methods and online submission information, please ensure that you have specified (1) whether consent was suitably informed and (2) what type you obtained (for instance, written or verbal). If your study included minors under age 18, state whether you obtained consent from parents or guardians. If the need for consent was waived by the ethics committee, please include this information.

5. Your ethics statement must appear in the Methods section of your manuscript. If your ethics statement is written in any section besides the Methods, please move it to the Methods section and delete it from any other section. Please also ensure that your ethics statement is included in your manuscript, as the ethics section of your online submission will not be published alongside your manuscript.

Reviewers' comments:

Reviewer's Responses to Questions

**Comments to the Author**

1. Is the manuscript technically sound, and do the data support the conclusions?

Reviewer #1: Partly

2. Has the statistical analysis been performed appropriately and rigorously? 

Reviewer #1: Yes

3. Have the authors made all data underlying the findings in their manuscript fully available?

Reviewer #1: Yes

4. Is the manuscript presented in an intelligible fashion and written in standard English?

Reviewer #1: Yes

5. Review Comments to the Author

Reviewer #1: The authors studied in a retrospective study a comparison between HG and CG in PPP for hemodynamically unstable pelvic fractures. They utilized propensity matching technique.

The co-primary outcomes were the rate of occurrence of postoperative wound infections and hemorrhage induced mortality (does not exactly reflect the title)

They demonstrated no differences in wound infections between the two groups, although the number of patients was too small to really conclude this (by the way-high rate of infections…). Beside the need for a larger N, since the focus of the study id wound infection, a detail of the organisms involved would be useful.

Cardiac arrest in the ER was included- were these patients successfully resuscitated and taken to the OR?

12 patients received angiography and angioembolization? What was the distribution between the groups?

Is it your institution practice to use either HG or CG at surgeon discretion? Since this is a retrospective study one cannot determine if the choice of treatment was based on severity of patient condition therefore may introduce a treatment bias.

Figure 2, I would recommend to delete.

In general, absent a protocol/guideline on when to use PPP vs angio etc.. it is difficult to conduct a retrospective study and in addition to a very small cohort.

6. PLOS authors have the option to publish the peer review history of their article (what does this mean?). If published, this will include your full peer review and any attached files.

Reviewer #1: No

---

## [Author Response · Author response to Decision Letter 0]

17 Apr 2020

Dear. Editor in chief and review. 

Thank you for your comments. I trust that this manuscript will be more fruitful by your comments. We really appreciate it. We revised this manuscript point to point. Thank you. 

The authors are required to respond to the reviewer's comments and to add a brief about their standard of care of patients with pelvic ring injury associated with haemodnamic instability. This will explain the use of HG in the study group and the role of angio-embolization.

 Thank you for your comment. It's good opinion. I appreciate it. According to our guideline for pelvic bone fracture with hemodynamically unstable based on the Rocky Mountain Regional Trauma Center at Denver Health, we perform PPP and/or EF and if bleeding is ongoing, perform pelvic angiography, which is 2ndary following PPP. I explained it little bit more in method section. In addition, there is no criteria for the use of hemostatic gauze and it was determined by surgeons. I add it in method section. 

Although the authors discussed the small sample size as a limitation of the study, they did not mention the cause of including only 53 out of 75 patients with PPP.

 Yes. Actually, small sample size is a limitation of this study. Among 75 patients, we excluded 22 patients. Because PPP didn't undergo for these 22 patients. We explained that in patient selection and data collection section before. I highlighted it sky blue color. 

-> We did it according to your comment. Thanks. 

2. Please provide additional details regarding participant consent. In the ethics statement in the Methods and online submission information, please ensure that you have specified (1) whether consent was suitably informed and (2) what type you obtained (for instance, written or verbal). If your study included minors under age 18, state whether you obtained consent from parents or guardians. If the need for consent was waived by the ethics committee, please include this information.

-> We did it accordiing to your comment. Thanks. 

a.Please clarify the sources of funding (financial or material support) for your study. List the grants or organizations that supported your study, including funding received from your institution.

b.State what role the funders took in the study. If the funders had no role in your study, please state: “The funders had no role in study design, data collection and analysis, decision to publish, or preparation of the manuscript.”

c.If any authors received a salary from any of your funders, please state which authors and which funders.

d.If you did not receive any funding for this study, please state: “The authors received no specific funding for this work.”

-> I put this comment "The authors received no specific funding for this work."

5. Review Comments to the Author

Reviewer #1: The authors studied in a retrospective study a comparison between HG and CG in PPP for hemodynamically unstable pelvic fractures. They utilized propensity matching technique.

The co-primary outcomes were the rate of occurrence of postoperative wound infections and hemorrhage induced mortality (does not exactly reflect the title)

 We changed the study title to ‘Safety and effectiveness of Kaolin-impregnated hemostatic gauze use in preperitoneal pelvic packing for patients with pelvic fractures and hemodynamic instability: propensity score matching analysis’. 

They demonstrated no differences in wound infections between the two groups, although the number of patients was too small to really conclude this (by the way-high rate of infections…). Beside the need for a larger N, since the focus of the study id wound infection, a detail of the organisms involved would be useful.

-> Thanks for comments. Therefore, I add this sentence. "Identified microorganisms in the HG group were Staphylococcus epidermidis, Staphylococcus epidermidis and Enterobacter aerogenes for one patient, and Enterococcus faecalis. Staphylococcus epidermidis, E.coli, and Methicillin resistent Staphylococcus capitis were identified in the CG." 

Cardiac arrest in the ER was included- were these patients successfully resuscitated and taken to the OR?

 Yes. We included those who were successfully resuscitated in the ER. 

12 patients received angiography and angioembolization? What was the distribution between the groups?

 According to the protocol, secondary pelvic angiography following PPP was performed in cases of ongoing bleeding even if PPP was performed. Embolization procedures were performed in cases of contrast media extravasations during pelvic angiography. Pelvic angiography was performed for 5 patients in the HG group and 4 patients in the CG (table 4). Pelvic angioembolization was performed for 2 patients in the CG, and 1 patient in the HG group (out of table). 

Is it your institution practice to use either HG or CG at surgeon discretion? Since this is a retrospective study one cannot determine if the choice of treatment was based on severity of patient condition therefore may introduce a treatment bias.

 Yes. When performing PPP, there were no criteria for the use of hemostatic gauze and it was determined by surgeon. But, it wasn't based on severity. The usage of hemostatic gauze was used randomly. We used propensity scoring matching, because there are some differences of characteristics in both groups which is definitely limitation of retrospective study. 

Figure 2, I would recommend to delete.

 Thank you for the comment. I will do it. 

In general, absent a protocol/guideline on when to use PPP vs angio etc.. it is difficult to conduct a retrospective study and in addition to a very small cohort.

 Thanks for your comment. Actually we have protocol. I mentioned it in material and method section. 

The management protocol for pelvic fractures with hemodynamically unstable was initiated in our trauma center in May 2014 according to management protocols created by the Rocky Mountain Regional Trauma Center at Denver Health (12). 

According to the protocol, secondary pelvic angiography following PPP was performed in cases of ongoing bleeding even if PPP was performed. Embolization procedures were performed in cases of contrast media extravasations during pelvic angiography. In cases requiring external fixator (EF) application, pelvic binders were not re-applied; rather, these were re-applied just after PPP, but only in cases without EF use. Pelvic binders were removed after patients became hemodynamically stable. 

Thank you again. We hope that you can keep safe in the situation. 

Sincerely,

Ji Young Jang, M.D.

Department of Surgery, Trauma Center, National Health Insurance Service Ilsan Hospital 

100 Ilsan-ro, Ilsan-donggu, Goyang-si, Gyenggi-do, Republic of Korea

Tel: +82-31-900-3624

E-mail: drjangjiyoung@gmail.com, jyjang@hanmail.net

---

## [Decision Letter · Decision Letter 1]

8 Jun 2020

PONE-D-19-34531R1

Safety and effectiveness of kaolin-impregnated hemostatic gauze use in preperitoneal pelvic packing for patients with pelvic fractures and hemodynamic instability: propensity score matching analysis

PLOS ONE

Dear Dr. Jang,

Thank you for submitting your manuscript to PLOS ONE. After careful consideration, we feel that it has merit but does not fully meet PLOS ONE’s publication criteria as it currently stands. Therefore, we invite you to submit a revised version of the manuscript that addresses the points raised during the review process.

We look forward to receiving your revised manuscript.

Kind regards,

Osama Farouk

Academic Editor

PLOS ONE

Reviewers' comments:

Reviewer's Responses to Questions

**Comments to the Author**

1. If the authors have adequately addressed your comments raised in a previous round of review and you feel that this manuscript is now acceptable for publication, you may indicate that here to bypass the “Comments to the Author” section, enter your conflict of interest statement in the “Confidential to Editor” section, and submit your "Accept" recommendation.

Reviewer #1: (No Response)

Reviewer #2: (No Response)

2. Is the manuscript technically sound, and do the data support the conclusions?

Reviewer #1: Partly

Reviewer #2: Partly

3. Has the statistical analysis been performed appropriately and rigorously? 

Reviewer #1: I Don't Know

Reviewer #2: Yes

4. Have the authors made all data underlying the findings in their manuscript fully available?

Reviewer #1: No

Reviewer #2: No

5. Is the manuscript presented in an intelligible fashion and written in standard English?

Reviewer #1: No

Reviewer #2: Yes

6. Review Comments to the Author

Reviewer #1: Although the authors modified the title to reflect the subject of their results, the primary outcome remains wound infections and as such this manuscript does not discuss or present adequate data on infections. For example a table on microorganisms, antibiotics, localized infection vs sepsis etc...

Again the size of this study is a major limitation.

Reviewer #2: I would like to thank the authors for their article and the valuable information. I have the following comments and questions.

I think that the major concerns from the readers side about your work would be:

1. Lines 90 - 101: The primary aim of your study was to evaluate the safety and effectiveness of kaolin-impregnated hemostatic gauze in PPP for patients with hemodynamic instability due to severe pelvic fractures. Between lines 90 - 101, you were describing your protocol for the management of patients with hemodynamic instability and pelvic fractures. You described the use of pelvic binders and external fixation according to the preference of the orthopaedic surgeons in the trauma team. When you analysed the results of kaolin-impregnated hemostatic gauze versus the control group PPP, we can't find data about how many patients had external fixation in each arm and how many patients had only pelvic binders. I think this is a very important piece of information to see if there any significant difference in the rates of external fixation vs pelvic binders in each treatment group. Furthermore, we can't find any data about the types or technique of external fixator, if it was C-clamp, supra-acetabular anterior external fixation, etc.

2. Lines 150 - 161: Regarding the patients' characteristics in your study, there were 17 patients LC-II (32%) and one patient LC-I. Hemodynamic instability with pelvic fractures are known to be significantly prevalent in pelvic fractures APC-III, LC-III and VS according to the data published by Young and Burgess in 1990 and according to our personal experience. However in your patient group, you have more patients with LC-II injuries compared to VS (30%) and APC-III (7.5%). Can you please explain this?

Other comments:

3. Line 71: Your study is a retrospective analysis or prospectively collected data and according to the patients and methods (Because the data were analyzed anonymously, informed consent was exempted.), however in line 71, you state in your inclusion criteria (3. agree to the collection and use of their medical information.). If the informed consent was exempted.

4. Lines 195 -198: I think details of identified micro-organisms in both treatment groups can be removed. Most important was the rates of deep infection.

7. PLOS authors have the option to publish the peer review history of their article (what does this mean?). If published, this will include your full peer review and any attached files.

Reviewer #1: No

Reviewer #2: Yes: Mohamed Kenawey

---

## [Author Response · Author response to Decision Letter 1]

30 Jun 2020

Dear. Editor in chief and reviewers. 

Thank you for your comments. I trust that this manuscript will be more fruitful by your comments. We really appreciate it. We revised this manuscript point to point. Thank you. 

Reviewer #1: Although the authors modified the title to reflect the subject of their results, the primary outcome remains wound infections and as such this manuscript does not discuss or present adequate data on infections. For example a table on microorganisms, antibiotics, localized infection vs sepsis etc...

Again the size of this study is a major limitation.

 We agree with the reviewer’s opinion and exclude wound infection from the primary outcome. Accordingly, the contents related to wound infection were removed from the manuscript and ‘safety’ was also excluded from the title of the study. 

In addition, I fully agree with the reviewer’s comments on the small study population. However, in this study, propensity score matching was performed to compensate for the problems of the small sized retrospective study, and the normality test of the two groups was checked to confirm the normal distribution of the two groups to confirm the statistical adequacy. So, we think our paper is unique, as there are no previous studies on the use of hemostatic gauze in patients with PPP. The results of this study are also meaningful in that they present on hypothesis for future randomized controlled studies. Through this, I think we can expect that a better hemostatic effect in patients undergoing PPP.

Thank you very much for your important review. 

Reviewer #2: I would like to thank the authors for their article and the valuable information. I have the following comments and questions.

I think that the major concerns from the readers side about your work would be:

1. Lines 90 - 101: The primary aim of your study was to evaluate the safety and effectiveness of kaolin-impregnated hemostatic gauze in PPP for patients with hemodynamic instability due to severe pelvic fractures. Between lines 90 - 101, you were describing your protocol for the management of patients with hemodynamic instability and pelvic fractures. You described the use of pelvic binders and external fixation according to the preference of the orthopaedic surgeons in the trauma team. When you analysed the results of kaolin-impregnated hemostatic gauze versus the control group PPP, we can't find data about how many patients had external fixation in each arm and how many patients had only pelvic binders. I think this is a very important piece of information to see if there any significant difference in the rates of external fixation vs pelvic binders in each treatment group. Furthermore, we can't find any data about the types or technique of external fixator, if it was C-clamp, supra-acetabular anterior external fixation, etc.



 Although orthopedic surgeons in our hospital decided whether or not to perform external fixation, it was performed in only 18.9% of unstable patients due to the lack of trauma-dedicated orthopedic surgeon in Korea. Thus, our protocol includes the application of the pelvic binder in the emergency room in all pelvic fracture patients with shock. External fixation application rates in both groups were added to the manuscript (Table 1,2,4). All external fixation was supra-acetabular external fixation and C-Clamp was not used in our trauma center. We added this contents to the material and method section. 

2. Lines 150 - 161: Regarding the patients' characteristics in your study, there were 17 patients LC-II (32%) and one patient LC-I. Hemodynamic instability with pelvic fractures are known to be significantly prevalent in pelvic fractures APC-III, LC-III and VS according to the data published by Young and Burgess in 1990 and according to our personal experience. However in your patient group, you have more patients with LC-II injuries compared to VS (30%) and APC-III (7.5%). Can you please explain this?

 Authors agree with the reviewer’s opinion. There are several reasons for the high proportion of LCII in our cohort and relatively low proportion of APCII and APCIII. 

First, because the trauma system in our area has not fully organized, 68% (36/53) of patients have been transferred from other hospitals to our trauma center, and the mean time from accident to ER arrival was 156 minutes, which is quite long. We think that Hemodynamically unstable patients with more severe pelvic fracture such as APC III or VS died prior to hospital arrival. 

Second, in our cohort, the pedestrian TA was 41.5% and the average age of patients was 60, which is relatively high compared to other cohorts. For these reasons, pelvic fracture caused by low energy injury often cause shock. These also seems to be the reason for the high proportion of LCII pelvic fracture patients in our cohort. 

Other comments:

3. Line 71: Your study is a retrospective analysis or prospectively collected data and according to the patients and methods (Because the data were analyzed anonymously, informed consent was exempted.), however in line 71, you state in your inclusion criteria (3. agree to the collection and use of their medical information.). If the informed consent was exempted.

 Third inclusion criterion was wrong, so we deleted it. 

4. Lines 195 -198: I think details of identified micro-organisms in both treatment groups can be removed. Most important was the rates of deep infection.

 Yes, I agree with the reviewer’s comments and have deleted this. 

Thank you for your thoughtful review.

---

## [Decision Letter · Decision Letter 2]

13 Jul 2020

Effectiveness of kaolin-impregnated hemostatic gauze use in preperitoneal pelvic packing for patients with pelvic fractures and hemodynamic instability: propensity score matching analysis

PONE-D-19-34531R2

Dear Dr. Jang,

We’re pleased to inform you that your manuscript has been judged scientifically suitable for publication and will be formally accepted for publication once it meets all outstanding technical requirements.

Kind regards,

Osama Farouk

Academic Editor

PLOS ONE

Additional Editor Comments (optional):

Reviewers' comments:

Reviewer's Responses to Questions

**Comments to the Author**

1. If the authors have adequately addressed your comments raised in a previous round of review and you feel that this manuscript is now acceptable for publication, you may indicate that here to bypass the “Comments to the Author” section, enter your conflict of interest statement in the “Confidential to Editor” section, and submit your "Accept" recommendation.

Reviewer #1: All comments have been addressed

Reviewer #2: All comments have been addressed

2. Is the manuscript technically sound, and do the data support the conclusions?

Reviewer #1: (No Response)

Reviewer #2: Yes

3. Has the statistical analysis been performed appropriately and rigorously? 

Reviewer #1: (No Response)

Reviewer #2: Yes

4. Have the authors made all data underlying the findings in their manuscript fully available?

Reviewer #1: (No Response)

Reviewer #2: No

5. Is the manuscript presented in an intelligible fashion and written in standard English?

Reviewer #1: (No Response)

Reviewer #2: Yes

6. Review Comments to the Author

Reviewer #1: (No Response)

Reviewer #2: (No Response)

7. PLOS authors have the option to publish the peer review history of their article (what does this mean?). If published, this will include your full peer review and any attached files.

Reviewer #1: No

Reviewer #2: No

---

## [Editor Report · Acceptance letter]

15 Jul 2020

PONE-D-19-34531R2 

Effectiveness of kaolin-impregnated hemostatic gauze use in preperitoneal pelvic packing for patients with pelvic fractures and hemodynamic instability: a propensity score matching analysis 

Dear Dr. Jang:

I'm pleased to inform you that your manuscript has been deemed suitable for publication in PLOS ONE. Congratulations! Your manuscript is now with our production department. 

Kind regards, 

on behalf of

Dr. Osama Farouk 

Academic Editor

PLOS ONE